

# Macroevolutionary patterns in intragenomic rDNA variability among planktonic foraminifera

Mattia Greco[1,2,3], Raphaël Morard[2], Kate Darling[4,5] and Michal Kucera[2]

[1] Institute of Oceanology, Polish Academy of Sciences, Sopot, Poland
[2] MARUM-Center for Marine Environmental Sciences, University of Bremen, Bremen, Germany
[3] Institut de Ciències del Mar (ICM), Consejo Superior de Investigaciones Científicas, Barcelona, Spain
[4] School of Geosciences, University of Edinburgh, Edinburgh, United Kingdom
[5] Biological and Environmental Sciences, University of Stirling, Stirling, United Kingdom

## ABSTRACT

Ribosomal intragenomic variability in prokaryotes and eukaryotes is a genomic feature commonly studied for its inflationary impact on molecular diversity assessments. However, the evolutionary mechanisms and distribution of this phenomenon within a microbial group are rarely explored. Here, we investigate the intragenomic variability in 33 species of planktonic foraminifera, calcifying marine protists, by inspecting 2,403 partial SSU sequences obtained from single-cell clone libraries. Our analyses show that polymorphisms are common among planktonic foraminifera species, but the number of polymorphic sites significantly differs among clades. With our molecular simulations, we could assess that most of these mutations are located in paired regions that do not affect the secondary structure of the SSU fragment. Finally, by mapping the number of polymorphic sites on the phylogeny of the clades, we were able to discuss the evolution and potential sources of intragenomic variability in planktonic foraminifera, linking this trait to the distinctive nuclear and genomic dynamics of this microbial group.

## INTRODUCTION

Ribosomal genes (rDNA) play a central role in cellular molecular processes and are present in multiple copies throughout eukaryotic and prokaryotic genomes (*Weider et al., 2005*; *Espejo & Plaza, 2018*). The occurrence of multiple copies in the genome is tied to the high demand for ribosomes, the organelles into which the expressed rRNA assembles, making rDNA the most transcribed genome locus (*Neidhart, Ingraham & Schaechter, 1990*; *Kobayashi, 2014*). Although high levels of transcription are well-known mutagenic sources (*Kim & Jinks-Robertson, 2012*), the rDNA gene sequences exhibit a striking homogeneity within a genome. This uniformity is maintained through evolutionary time within a species, resulting from the molecular process of concerted evolution (*Elder & Turner, 1995*; *Espejo & Plaza, 2018*), but divergence becomes evident when rDNA sequences from different species are compared. The process preventing divergence among the multiple

Corresponding author
Mattia Greco,
mattia_greco@outlook.com

gene copies within a single genome is gene homogenization, where related genes within a species undergo genetic exchange (*Dover, 1982*) and the molecular mechanisms that induce it (gene conversion, gene amplification and unequal crossing-over) have been identified in both eukaryotes and prokaryotes (*Dover, 1982*; *Liao, 2000*). However, multiple factors can affect the homogenization of ribosomal genes which interfere with concerted evolution, resulting in intragenomic variability appearing among rDNA copies within an organism. This phenomenon is particularly common in microbial organisms, where rDNA intragenomic variability has been linked to the presence of a particularly large number of gene copies per cell (*Gong et al., 2013*), the displacement of rDNA gene copies among different nuclei (*Lanfranco, Delpero & Bonfante, 1999*), the presence of non-coding rDNA copies, *i.e.*, pseudogenes (*Gribble & Anderson, 2007*), and inter-specific hybridization (*Boucher et al., 2004*; *Pillet, Fontaine & Pawlowski, 2012*).

Since ribosomal genes are widely used markers for metabarcoding surveys, research on rDNA intragenomic variability has been focused on its biasing effects on microbial diversity or phylogenetic assessments (*Thornhill, Lajeunesse & Santos, 2007*; *Sun et al., 2013*; *Zhao et al., 2019*; *Hassler et al., 2022*; *Sandin, Romac & Not, 2022*). Despite the growing literature quantifying intragenomic variability in microbial species, knowledge of how this trait evolves across taxa is remarkably incomplete. Here we use the microbial group of the foraminifera as a case study since their intragenomic variability is widespread (*Weber & Pawlowski, 2014*). We particularly focus on the planktonic species. This group has a known evolutionary history that can be reconstructed from the fossil record (*Aze et al., 2011*) and their modest taxonomic diversity with about 50 extant species (*Brummer & Kucera, 2022*) is structured into three clades, the Spinose, the Non-Spinose and the Microperforate. Most importantly, because of extensive barcoding efforts, the diversity in the planktonic foraminifera group has a near-complete taxonomic coverage of the 3′ end of the SSU rDNA gene, with highly replicated sampling among the taxa (*Morard et al., 2015*). This comprehensive coverage enables us to systematically evaluate intragenomic variability throughout the group and assess the patterns and macroevolutionary mechanisms responsible for generating such variability. To this end, we expanded the resolution and scope of the existing data by amplifying long sequences using clone libraries, resulting in a collection of 2,403 partial SSU sequences. The data now encompasses five hypervariable regions (or helices) from 33 species of planktonic foraminifera distributed across the major clades of the group. We used these sequences to (i) assess the incidence of intragenomic polymorphism in each species, (ii) examine its effects on the secondary structure of the rDNA molecule, and (iii) explore the evolutionary patterns of the observed incidence and degree of sequence variability in the context of planktonic foraminifera evolutionary history as reconstructed from the fossil record.

## MATERIALS AND METHODS

### Collection

Living planktonic foraminifera specimens identified as *Globigerina bulloides, Globorotalia eastropacia, Neogloboquadrina dutertrei, Globigerinoides elongatus, Turborotalita humilis, Neogloboquadrina incompta, Globorotalia inflata, Pulleniatina obliquiloculata,*

*Neogloboquadrina pachyderma, Hastigerina pelagica, Turborotalita quinqueloba, Trilobatus sacculifer, Globorotalia scitula, Globorotalia truncatulinoides, Globorotalia ungulata*, and *Orbulina universa* were sampled between 1997 and 2017 during the cruises M37/2a, P247, ARK XV/I, ARK XV/II, JR 48, ANT XVIII/56, CD159, D286, P334, M74-1b, 64PE304, Iberia-Forams, SO226, MSM39, MSM44, M113-2, M124 and M133, during transects on the vessels Prof Logachev (August 1997), RV Welwitschia (November 2001), and Sir Wilfrid Laurier (July 2002), as well as during a near-shore collection in the Santa Barbara Channel in 1999 and Santa Catalina Island in 2015. The specimens were recovered by stratified net sampling, simple nets with mesh size above 100 μm, or SCUBA collection.

Living specimens were picked from the plankton and either taxonomically identified, cleaned and directly isolated into extraction buffer or transferred onto cardboard slides, air-dried and stored at −20 °C (*Weiner et al., 2016*). The air-dried specimens were then taxonomically identified under a stereomicroscope before isolation into the buffer in the laboratory.

## DNA extraction, amplification and sequencing

For the newly collected specimens, DNA extraction was performed using either DOC (Sodium-deoxycholate), urea buffer, or a GITC* (Guanidinium isothiocyanate) protocol (*Weiner et al., 2016*). New sequences were then obtained by amplification of a fragment located at the 3′ end of the SSU (~1,000 bp) using the primer pairs S14F1-1528R or S14p-1528R (*Weiner et al., 2016*). PCRs were performed using the polymerase PHUSION (Thermo Fisher Scientific, Waltham, MA, USA) following manufacturer instructions. The PCR products obtained were purified using the QIAquick PCR purification Kit (QIAGEN, Hilden, Germany) and directly sequenced by an external provider (LGC Genomics, Berlin, Germany).

To assess the extent of intragenomic variability, 77 specimens belonging to 12 species were cloned using the Zero Blunt TOPO PCR cloning Kit (Invitrogen, Waltham, MA, USA) with TOP10 chemically competent cells following the manufacturer's instructions. Between 1 and 42 clones were sequenced per individual. The chromatograms were carefully checked and only sequences of sufficient quality were kept.

## Dataset assembly, sequence partitioning and quality check

The newly generated sequences were curated following the standards of the *Planktonic foraminifera Ribosomal Reference* database-PFR² framework (*Morard et al., 2015*). Briefly, all the sequences were manually aligned with SEAVIEW 4.0 (*Gouy, Guindon & Gascuel, 2010*) to the borders of each variable region. Successively, all the new sequences were partitioned into six conserved (32–37, 37–41, 41–43, 43–44, 47–49 and 50) and five variable (37f, 41f, 43e, 45e–47f and 49e) regions (*Pawlowski & Lecroq, 2010*). The partitioned sequences were then complemented with planktonic foraminifera SSU sequences with curated taxonomy extracted from PFR² version 1.0 (*Morard et al., 2015*) as well as with all sequences published after its release. The resulting dataset included 6,795 partial SSU sequences from 43 planktonic foraminifera species.

For our investigation of intragenomic variability of planktonic foraminifera, we focussed on the five variable regions of the foraminiferal SSU (37f, 41f, 43e, 45-e–47f and 49e). As a quality check of the individual chromatograms of the publicly available sequences was not possible, we performed a further filtering step at the level of single variable regions to exclude rare variants or potential sequence artifacts. For this purpose, we retained only variable region sequences that were fully covered and observed at least twice in our dataset. Finally, only species represented by at least two specimens in our compiled dataset were considered for the analyses and we removed sequenced that could not be reliably assigned to a single specimen voucher.

The final dataset included 2,403 clonal sequences from 33 planktonic foraminifera species and 4,073 sequences obtained by direct sequencing (See Table S1).

In order to discuss our results in the light of previous intragenomic estimates on foraminifera, we also included in our analyses data from the benthic species *Cassidulina laevigata*, *Oridorsalis umbonatus*, and *Ammonia* sp. presented in *Weber & Pawlowski (2014)*. To be fully comparable, the benthic sequences were also partitioned into hypervariable regions but, given the shorter length, they included only the 37f, 41f, and 43e helices.

## Assessment of intragenomic variability

Intragenomic polymorphism was assessed for each specimen and variable region. As a first step, we aligned all the sequences obtained from the same specimen and covering the same variable region using MAFFT v.7 (*Katoh et al., 2002*). We then annotated the position and the number of the polymorphic site/s in each alignment (*i.e.*, specimen) using the function seg.sites in the R package ape (*Paradis, Claude & Strimmer, 2004*; *R Core Team, 2017*).

To test for differences in the overall number of polymorphic sites among clades, we performed a Kruskal-Wallis test. This is a non-parametric analysis that allows us to determine if there are statistically significant differences between two or more groups. Then, we carried out a Dunn test to perform pairwise comparisons between the s different planktonic foraminifera clades. The analyses were carried out using the R package ggstatsplot (*Patil, 2021*).

The occurrence (presence/absence) of polymorphism in each species and variable region was used to calculate the incidence of intragenomic variability as follows:

*Incidence = (N specimens showing polymorphism/N specimens) * 100*

An overall incidence was also calculated using the same formula but considering the occurrence of polymorphisms in a specimen across all the hypervariable regions analysed.

Since these two metrics are largely dependent on the sample size, we also derived the cloning effort for each species as follows:

$$Cloning\ effort = \left( \frac{N\ specimens\ cloned}{N\ specimens\ cloned} + N\ specimens\ direclty\ sequenced \right) * 100$$

## Secondary structure analyses

To map the position of each polymorphic site on the secondary structure of each of the rRNA fragments analysed, we derived a consensus sequence from each specimen alignment using the consensus function from the seqinR package (*Charif & Lobry, 2007*). The secondary structures of the sequences obtained were then annotated in the Vienna RNA dot-bracket format (*Washietl, Hofacker & Stadler, 2005*) using the RNA fold web server (http://rna.tbi.univie.ac.at/cgi-bin/RNAWebSuite/RNAfold.cgi) (*Hofacker, 2003*) with default parameters. The drawings of the secondary structure obtained were visualised with VARNA v. 3.93 (*Darty, Alain & Ponty, 2009*).

Next, using a custom R script, we mapped the polymorphism position (see the previous section) onto the molecule structure notation. To investigate the potential effects of the mutations on the molecular structure of the fragment analysed we recorded their occurrence on stem or loop sites. For each species, counts on polymorphism locations on the secondary structure were then used to derive Stem-Loop ratio (SLr) defined as:

$$SLr = N \text{ polymorphic sites on } Stem / N \text{ polymorphic sites on } Loop$$

Differences in SLr among clades were tested by performing a Kruskal-Wallis followed by a Dunn test using the R package ggstatsplot (*Patil, 2021*) as described in the previous section.

## Phylogenetic analyses

To test the evolutionary component of intragenomic variability among planktonic foraminifera, we recovered the phylogenetic history of the Macroperforate group (Spinose and Non-Spinose clades) based on fossil evidence as presented in *Aze et al. (2011)*. The original tree including modern and extinct lineages was imported in R using the package paleoPhylo (*Ezard & Purvis, 2009*) and pruned to match our sampling using the drop.tip function implemented in the R package ape (*Paradis, Claude & Strimmer, 2004*). The resulting phylogeny was complemented with the species *N. incompta* and *G. elongatus*, which were added based on their First Appearance Datum (*Darling, Kucera & Wade, 2007*; *Aurahs et al., 2011*) using the bind.tip function in the R package ape (*Paradis, Claude & Strimmer, 2004*). For the Microperforate clade, we used a molecular-based phylogenetic tree (*Morard et al., 2019*). The tree was non-ultrametric and we, therefore, transformed it into an ultrametric tree with the function chronos in the ape R package (*Paradis, Claude & Strimmer, 2004*). It was then pruned following the procedure described above.

To test the presence of a phylogenetic signal, we calculated the median number of polymorphic sites (Mps), the average number of polymorphic sites (Aps) and the overall incidence of intragenomic variability (Iniv). We then performed a Abouheif's Cmean phylogenetic test (*Abouheif, 1999*) for each of the variables using the Phylosignal package in R (*Keck et al., 2016*). Variables showing significant results were then used to reconstruct the Ancestral State of the group using the function FastAnc as implemented in the R package phytools (*Revell, 2012*).
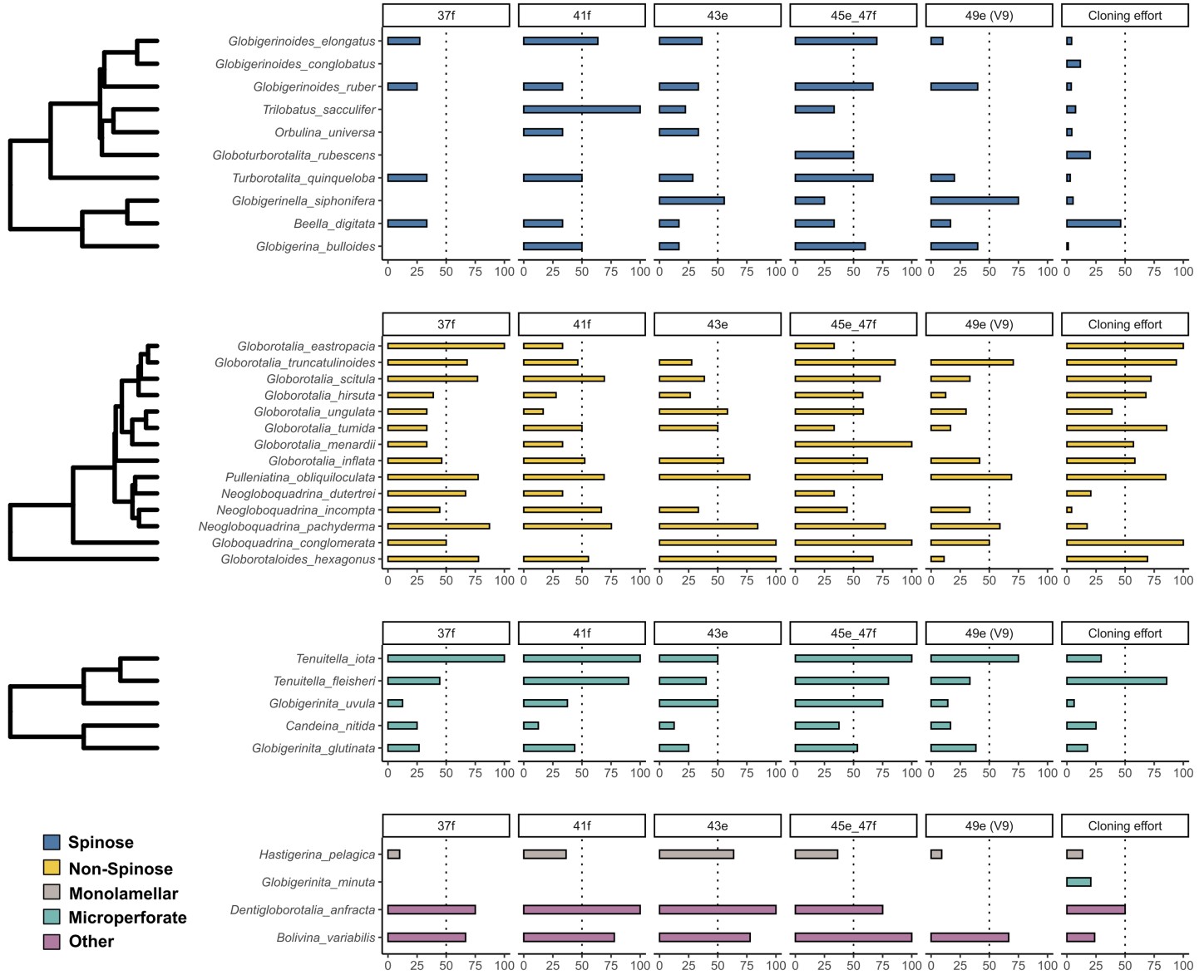

**Figure 1** **Incidence of intragenomic variability in planktonic foraminifera species.** The bar-plots indicate the percentage of specimens presenting intragenomic variability for each of the hypervariable regions considered in this study. Cloning effort represents the percentage of specimens cloned out of all the specimens that were sequenced. The bars are color-coded according to clades.

# RESULTS

The incidence of intragenomic variability varied greatly across the hypervariable regions and the species analysed (Fig. 1). On average, Non-Spinose species had a higher incidence, reaching levels beyond 50% for each of the helices studied. Within this clade, the maximum values were observed in *G. eastropacia* (100%, 37f), *N. pachyderma* (75%, 41f), *G. conglomerata* (100%, 43e and 45e–47f), *G. hexagonus* (100%, 43e), *G. menardii* (100%, 45e–47f) and *G. truncatulinoides* (70%, 49e). Conversely, the incidence was above 50% in only the three Spinose species *T. sacculifer* (100%, 41f), *G. elongatus* (70%, 45e–47f) and *G. siphonifera* (75%, 49e) and absent in *G. conglobatus*. Intragenomic variability was also
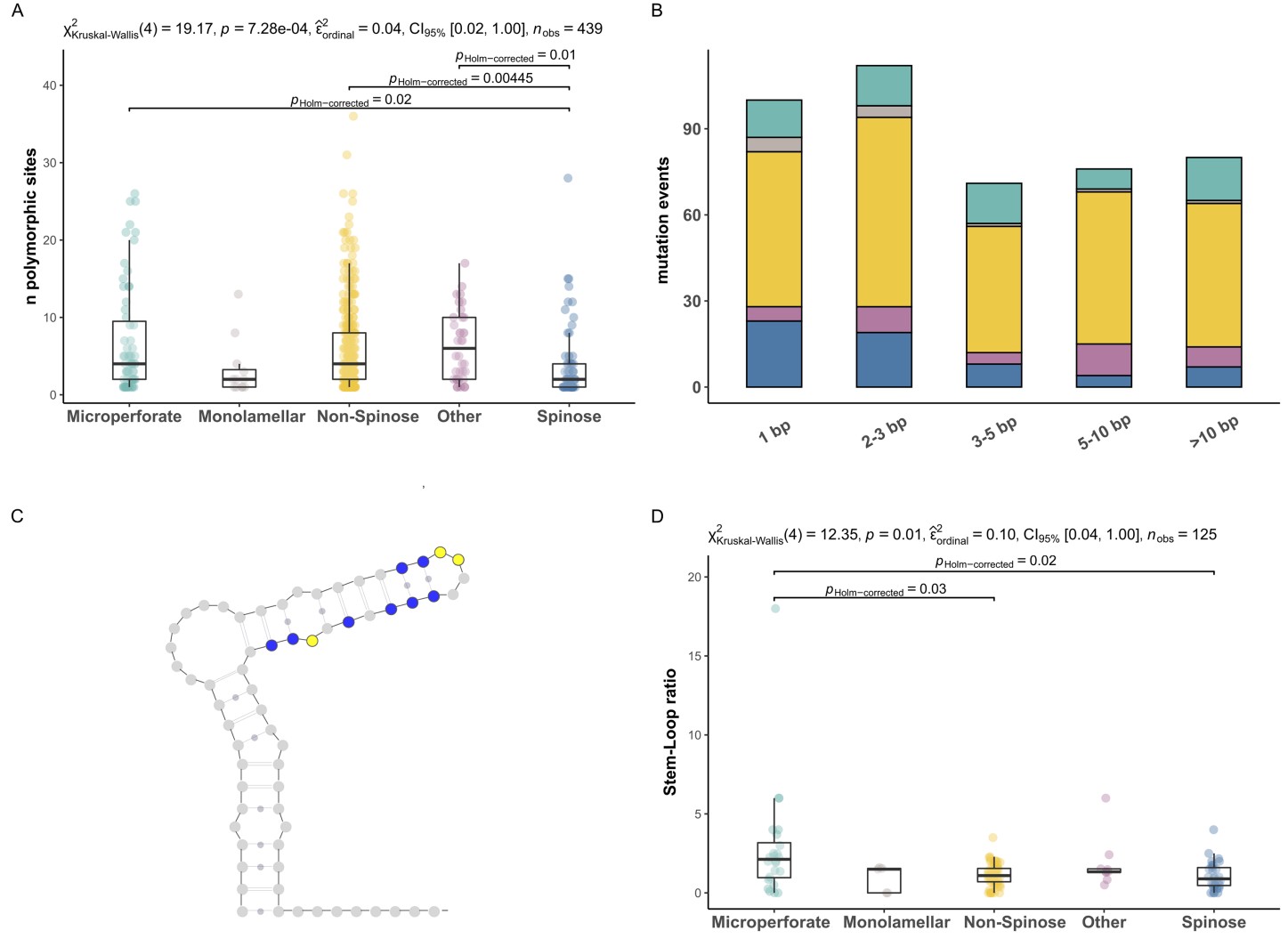

**Figure 2 Polymorphic sites and stem-loop ratio in planktonic foraminifera clades.** (A) Number of polymorphic sites in each clade. Statistically distinct group pairs are connected by lines indicating significance values. (B) Distribution of mutation events for each size class across clades, colour indicates clades as in Fig. 1. (C) Example of molecular simulation of the SSU secondary structure highlighting polymorphic sites located on stem (blue) and loop (yellow) regions. (D) Stem-Loop ratio derived for each clade. Statistically distinct group pairs are connected by lines indicating significance values.

prevalent among the Microperforate taxa (Fig. 1). Their average incidence was 53%, ranging between 13% and 100% except for *G. minuta*, which displayed no polymorphism.

To investigate the extent of the mutations causing the polymorphism, we proceeded to analyse the number of polymorphic sites and tested for differences between clades (Fig. 2). We found that the number of polymorphic sites varied between 1 and 36 base pairs. Remarkably, the number of times the presence of polymorphisms was detected in a specimen (*i.e.*, mutation events) was substantially higher in Non-Spinose species, irrespectively of the number of polymorphic sites ascribable to the specific mutation event (Fig. 2B). In addition, the species *D. anfracta* and *B. variabilis* (Other in Fig. 2A), exhibited a distribution centred towards a higher number of polymorphic sites (IQR = 8).

The Kruskal–Wallis test showed that the number of polymorphic sites statistically differed among the clades analysed ($p = 0.01$) (Fig. 2A). Specifically, the number of polymorphic sites was found to be significantly smaller in the Spinose clade compared to the Microperforate ($p = 0.02$), and Non-Spinose ($p = 0.004$) clades and in comparison with the species *D. anfracta* and *B. variabilis* ($p = 0.01$) as revealed by the paired Dunn test (Fig. 2).

To assess the effect of polymorphisms on the SSU secondary structure, we counted the number of mutations located in the Stem or Loop regions of the molecule to obtain the Stem/Loop ratio (SLr) (Figs. 2C and 2D) and tested for differences between clades. The SLr assumed values larger than 0 in 88% of the polymorphisms observed, indicating that a higher number of mutations were located in the stem regions in the vast majority of the specimens. The Kruskal–Wallis test showed that the SLr statistically differed among the clades analysed ($p = 0.01$) (Fig. 2D). In particular, the paired Dunn test (Fig. 2) indicated that the SLr was significantly higher in the Microperforate clade compared to the Spinose ($p = 0.02$), and Non-Spinose ($p = 0.03$) species.

We further investigated the macroevolutionary signal behind intragenomic variability in planktonic foraminifera by conducting the Abouheif's $C_{mean}$ phylogenetic test (*Abouheif, 1999*). We performed the analysis separately for each morphogroup, using the fossil phylogeny for Spinose and Non-Spinose (*Aze et al., 2011*) and the molecular phylogeny for Microperforate species (*Morard et al., 2019*). The species-specific traits tested were: the median number of polymorphic sites (Mps), the average number of polymorphic sites (Aps) and the overall incidence of intragenomic variability (Iniv). A significant positive phylogenetic signal was retrieved in the Non-Spinose phylogeny for the Aps ($C_{mean} = 0.28$, $p = 0.02$) and Mps ($C_{mean} = 0.35$, $p = 0.01$) traits.

Finally, the trait with the strongest phylogenetic signal, Mps, was selected for the reconstruction of the ancestral state and plotted on the fossil phylogeny of each morphogroup. The median number of polymorphic sites varied within major clades but was on average higher in Non-Spinose (3.34) and Microperforate (4.73) than in Spinose (1.75). The highest Mps values were observed in the Spinose species *T. quinqueloba* (Mps = 8), the Non-Spinose species *N. pachyderma* (Mps = 8), *N. incompta* (Mps = 11) and the Microperforate species *T. iota* (Mps = 8.5). Importantly, the analysis suggested that some polymorphic sites (~3 base pairs) were present in the ancestor of the group and have been successively reduced in some lineages, mostly Spinose, but multiplied in others (Fig. 3).

A stark reduction for polymorphic sites was also evident when data from the benthic Rotaliida and planktonic clades were compared (Fig. 4). The boxplot in Fig. 4 shows that the Mps for benthic Rotaliida (Mps = 12) was three times larger than in Microperforates (Mps = 4), four times larger than in Non-Spinose (Mps = 3) and six times larger than in Spinose (Mps = 2) for the 37f, 41f and 43e helices.

## DISCUSSION

As previously shown for benthic foraminifera (*Holzmann, Piller & Pawlowski, 1996*; *Pillet, Fontaine & Pawlowski, 2012*; *Weber & Pawlowski, 2014*; *Girard et al., 2022*), the

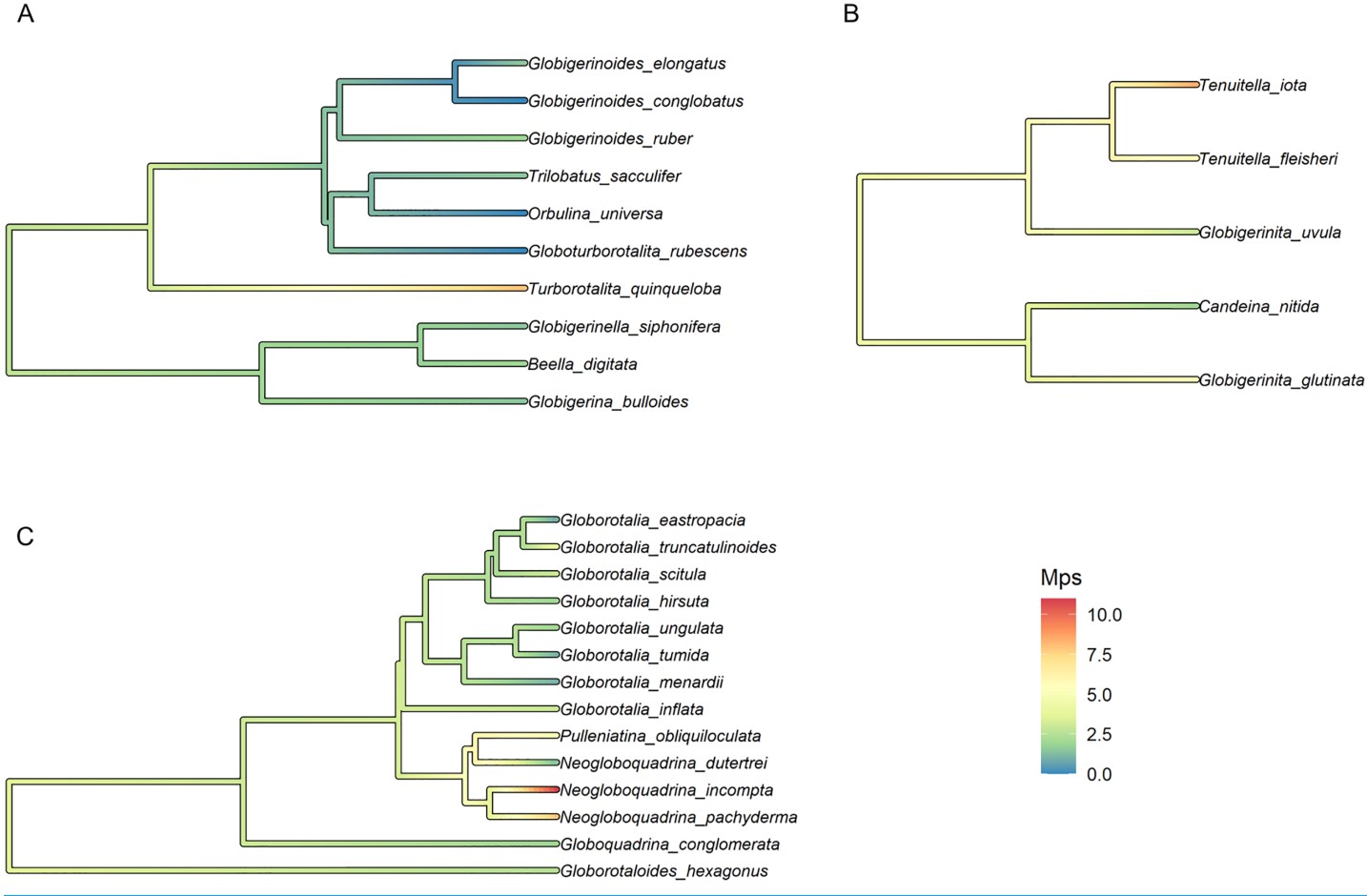

**Figure 3 Ancestral state reconstruction of the median number of polymorphic sites.** Ancestral state reconstruction of the median number of polymorphic sites (Mps) in (A) Spinose, (B) Microperforate and (C) Non-Spinose phylogenetic trees.

occurrence of intragenomic variability is also widespread among planktonic foraminifera (Fig. 1). The incidence of this trait is particularly high among the Non-Spinose and Microperforate species together with the closely related species *B. variabilis* and *D. anfracta*. Importantly, our analysis shows that intragenomic polymorphisms are also found on the V9 region in 24 of the species analysed, with incidence values above 60% in *G. truncatulinoides* and *G. siphonifera* (Fig. 1). The V9 is a widely used marker for assessing the diversity of microbial plankton in metabarcoding surveys (*e.g.*, *de Vargas et al., 2015*; *Piredda et al., 2017*), thus the extensive presence of intragenomic variability in this region calls for a careful interpretation of diversity estimates of planktonic foraminifera community based on this marker.

Admittedly, while providing a valid overview of the occurrence of intragenomic variability in planktonic foraminifera, the incidence is impacted by the different cloning efforts (sampling bias) for the species present in our compiled dataset (Fig. 1). Such disparity is likely the result of earlier observations that showed no intragenomic variability among Spinose lineages (*André et al., 2014*) and high levels among Non-Spinose and Microperforates (*Darling et al., 2006*; *Morard et al., 2016*, *2019*), which possibly skewed

later cloning effort towards species belonging to these two clades. Another factor contributing to this asymmetric effort is the good quality data resulting from the direct sequencing of Spinose specimens that, thus, do not require cloning for obtaining interpretable sequences.

Conversely, the number of polymorphic sites and SLr are robust measures, independent of the number of specimens analysed. Together, these two parameters can provide a verifiable framework to understand potential sources and mechanisms behind the extent and evolution of intragenomic variability in planktonic foraminifera ribosomal genes.

The first pattern emerging from our analyses is that the occurrence of polymorphic sites on the ribosomal genes is a common trait in every planktonic foraminifera clade (Figs. 1 and 2). A potential explanation for this result is that we included non-coding sequences in our analyses (*Gribble & Anderson, 2007*). However, our RNA molecule simulations suggest that it is unlikely that the sequences analysed are pseudogenes since most of the polymorphic sites are located on paired (stem) regions (Fig. 2D), indicating that their ribosomal secondary structure is subject to functional constraints (*Smit, Widmann & Knight, 2007*). Alternatively, the presence of a particularly large number of rDNA copies has also been proposed as a source of intragenomic polymorphisms in ciliates (*Gong et al., 2013*) and fungi (*Simon & Weiss, 2008*). A recent investigation using single-cell qPCR has shown that planktonic foraminifera carry between ~300 and 350,000 copies of the SSU genes, a trait shared among species of different clades (*Milivojevic et al., 2021*) and common also in benthic lineages (*Weber & Pawlowski, 2013*). Notably, in their study, *Milivojevic et al. (2021)*, show that the number of gene copies in planktonic foraminifera varies between different species and could reflect the ploidy status of the sequenced specimens.

Another genomic feature associated with intragenomic polymorphism is nuclear dualism or heterokaryosis. It has been proposed that, in ciliates, protists characterised by the presence of a diploid micronucleus and a polyploid macronucleus, polymorphic sites could result from rDNA amplification occurring during the development or amitotic divisions of the macronucleus (*Wang et al., 2019*). Importantly, canonical heterokaryosis has been also reported in several benthic foraminifera species within the order Rotaliida (reviewed in *Goetz et al. (2022)*), the clade from which modern planktonic lineages diversified from benthic ancestors (*Pawlowski, Holzmann & Tyszka, 2013*).

Finally, intragenomic polymorphism could be the result of interspecific hybridization as shown for some benthic species (*Pillet, Fontaine & Pawlowski, 2012*), however, given that no evidence of this phenomenon exists for planktonic foraminifera (*Weiner et al., 2014*), we can rule it out as a possible explanation of our results.

Together, these observations can help provide an interpretation for the widespread occurrence of intragenomic polymorphism in planktonic clades (Fig. 1) as well as its presence in the ancestral nodes of the group (Fig. 3): we speculate that this genomic feature could represent an evolutionary relict deriving from benthic ancestors characterised by nuclear dualism and/or disproportionately high rDNA gene copy number.

The high number of polymorphic sites among benthic species shown in Fig. 4 also suggests that intragenomic variability is a shared trait within the order Rotaliida and

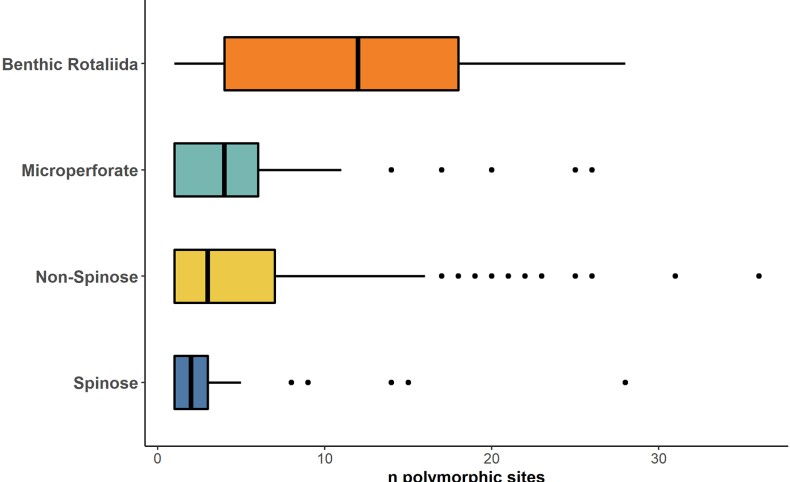

**Figure 4 Number of polymorphic sites in benthic and planktonic taxa.** Boxplot comparing the number of polymorphic sites detected on the 37f, 41f, and 43e helices between benthic rotaliida, Microperforate, Non-Spinose, and Spinose, color-coded by clade.

therefore, potentially present in the benthic ancestors of the planktonic clades. Furthermore, our analysis indicates a substantial reduction in the number of polymorphic sites in the planktonic taxa (Fig. 4). Interestingly, this pattern appears to trace the estimated time of diversification of the planktonic clades from the benthos (*Morard et al., 2022*). The Spinose clade is the most ancient and on average displays a lower number of polymorphic sites than the more recently diversified Microperforate clade which shows the highest (Fig. 4). This suggests that planktonic clades have gradually reduced their intragenomic variability through their evolution.

We should also note that our explanation potentially implies the presence of heterokaryosis in modern planktonic foraminifera, a hypothesis that would require further investigations employing laboratory techniques like DAPI staining combined with confocal microscopy to be confirmed (*e.g.*, *Bellec, Maurer-Alcala & Katz, 2014*).

The second pattern emerging from our analyses is that there are significantly fewer polymorphic sites present in the rDNA copies of the Spinose clade compared to those of the Microperforate and Non-Spinose clades. (Fig. 2A). This observation holds across the different magnitudes of mutation events observed, ruling out the possibility that the difference is simply the result of single polymorphic sites potentially ascribable to random errors introduced during the polymerase chain reaction amplification process (Fig. 2B).

The prevalence of intragenomic variability in Non-Spinose and Microperforates suggests the presence in these species of a mechanism that hinders the process of gene homogenization of the rDNA gene copies, amplifying the effects of the elevated gene copy number or nuclear dualism discussed above. The explanation might lie in the different life cycle strategies of some planktonic foraminifera species. Previous studies on ciliates and fungi have shown that in multinucleate organisms, intragenomic rDNA polymorphism is higher when multiple macronuclei or nuclei are present within the same cell or spore (*Lanfranco, Delpero & Bonfante, 1999*; *Zhao et al., 2019*). This is because the rDNA gene

copies emerging during genome multiplication in different (macro) nuclei undergo gene conversion within the same nucleus but not between all nuclei, originating polymorphic copies within the same specimen (*Lanfranco, Delpero & Bonfante, 1999*). Indeed, during the asexual generation, foraminifera are multinucleated while sexually reproducing individuals only possess a single nucleus (*Goetz et al., 2022*). To date, asexual reproduction has been only reported in Microperforate and Non-Spinose species (*Davis et al., 2020*; *Takagi, 2020*; *Meilland et al., 2022*), while Spinose species have only been observed to reproduce *via* gametogenesis (*i.e.*, sexually) (*e.g.*, *Bijma, Erez & Hemleben, 1990*). We speculate that the incidence of asexual reproduction in a lineage could drive the elevated number of polymorphic sites that we observe in these two clades. Although the prevalence of multinucleated asexually reproducing planktonic foraminifera needs further investigation, the significant phylogenetic signal of Mps and Aps recorded in the Non-Spinose group suggests that intragenomic polymorphism is a heritable trait, albeit likely due to a secondary correlation with a higher incidence of asexual reproduction within the clade.

## CONCLUSIONS

In this study, we investigated the occurrence and extent of intragenomic variability in planktonic foraminifera by analysing a compilation of 2,403 sequences from the clone libraries of 33 species. Our analysis shows that intra-individual mutations on SSU genes are common across all studied clades, but are more prevalent in Non-Spinose and Microperforate species. Furthermore, we identified the number of polymorphic sites as an important macroevolutionary trait, possibly reflecting the evolution of the nuclear and genomic architecture in the group. Our findings significantly advance our understanding of the molecular biology of planktonic foraminifera and underline the importance of studying genomic features in a phylogenetic context to gain insights into the evolutionary history of microbes.

### Funding

Laboratory work was supported by an Erasmus + Traineeship scholarship awarded to Mattia Greco by the University of Bologna. Kate Darling received funding from the Natural Environment Research Council (NERC) of the United Kingdom (grants NER/J/S/2000/00860 and NE/D009707/1) for this research. Raphaël Morard's work was funded by the Cluster of Excellence "The Ocean Floor—Earth's Uncharted Interface" (EXC-2077, Project 390741603) funded by the German Research Foundation (DFG). The funders had no role in study design, data collection and analysis, decision to publish, or preparation of the manuscript.

### Grant Disclosures

The following grant information was disclosed by the authors:
University of Bologna.

Natural Environment Research Council (NERC) of the United Kingdom: NER/J/S/2000/00860 and NE/D009707/1.

Cluster of Excellence "The Ocean Floor—Earth's Uncharted Interface": EXC-2077, Project 390741603.

German Research Foundation (DFG).

## Competing Interests

The authors declare that they have no competing interests.

## Author Contributions

- Mattia Greco conceived and designed the experiments, performed the experiments, analyzed the data, prepared figures and/or tables, authored or reviewed drafts of the article, and approved the final draft.
- Raphaël Morard conceived and designed the experiments, performed the experiments, authored or reviewed drafts of the article, and approved the final draft.
- Kate Darling conceived and designed the experiments, performed the experiments, authored or reviewed drafts of the article, and approved the final draft.
- Michal Kucera conceived and designed the experiments, authored or reviewed drafts of the article, and approved the final draft.

## Data Availability

The accession numbers of the sequences are available in Table S1.

The data tables and phylogenetic trees used for the analyses are available at FigShare: Greco, Mattia (2023): Supporting data to 'Macroevolutionary patterns in intragenomic rDNA variability among planktonic foraminifera'. figshare. Dataset. https://doi.org/10.6084/m9.figshare.21316374.v1.

The code used to generate the figures is available at GitHub and Zenodo: https://github.com/MatGreco90/IG_variability_planktonic_foraminifera.

Mattia Greco. (2023). MatGreco90/IG_variability_planktonic_foraminifera: zenodo_release (v1.0.0). Zenodo. https://doi.org/10.5281/zenodo.7797515.

## Supplemental Information

Supplemental information for this article can be found online at http://dx.doi.org/10.7717/peerj.15255#supplemental-information.

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
