# Peer review of "Macroevolutionary patterns in intragenomic rDNA variability among planktonic foraminifera"

_PeerJ, doi:10.7717/peerj.15255_

## Round 0.1 · original submission · Major Revisions

I have heard from two reviewers, both of whom have offered fair and constructive comments on your work. The comments appear (to me) to be straightforward and should be relatively easy to address; my decision of "Major revisions" is based more on the number of comments than anything else. I look forward to seeing a revised version of your work.

·

Basic reporting

The authors report on the intragenomic variability in planktonic foraminifera clades by comparing multiple clones of SSU rDNA from individual cells to direct sequencing products and publicly available sequences. A clear and concise introduction frames the hypotheses in this study, using recent relevant literature; however, in line 71 the authors reference an existing dataset which requires further clarification. What were the main findings from previous studies of planktonic foraminifera and how did these finding influence the manuscripts hypotheses?

Overall, the reporting of major findings is sufficient for publication with some minor revisions to the methods and discussion to improve clarity and summarize the impacts of the main findings.

Experimental design

The research presented here falls within the scope of PeerJ and addresses critical knowledge gaps in our understanding of microbial diversity, including the possible evolutionary origins of intragenomic variability.

The manuscript could be improved by addressing the following points:
1. Create a supplemental table describing the specimen ID, taxon ID, collection method, sampling location, and depth
2. Create variables for the equations used and present equations used in this study in standard format (L147-156, L171)
3. Share R scripts in a public repository

Validity of the findings

Here, the authors could improve the discussion of their findings by linking the patterns of polymorphisms to sequencing efforts and phylogenies more clearly by addressing the following points:

L199: Note the number of clones for each taxon in addition to the prevalence of polymorphisms
L252: Address the source of cloning bias, was this based on samples available or decided based on results of previous studies?
L256: I would disagree that the number of polymorphic sites is independent of the number of specimens sequenced, given not all taxa had 100% prevalence of polymorphisms for any given region in this study. Undersampling could potentially lead to biased detection of intragenomic variability.
L269: Were copy number variations correlated with life stages and/or taxonomy in the referenced study?

The paragraphs beginning at lines 272 and 283 have redundant themes and could be combined for clarity.

L293: The authors describe the varying lengths of the polymorphic regions, but discussion would benefit from relevant literature describing how/why these regions may emerge in foraminifera or other protists. Is there significance to the lengths of the mutation regions?

Line 310: The authors describe asexual reproduction in 2 of the three clades of planktonic foraminifera. Is there any literature describing reproduction of Spinose foraminifera? Does it compare or contrast with the other 2 clades?

The authors include the prevalence of intragenomic variability within the V9 region in Figure 1; however, the discussion should also include the V9 region findings within the context as a common marker for microbial community diversity

Additional comments

Minor changes

L65: Clarify what is meant by "allowing a replicated analysis of their evolutionary history"
L92: Arrange sampling locations chronologically
L93: Delete either
L96: Specify extraction buffer used
L100: Define acronyms for DOC and GITC*
L101: Edit sentence beginning with "Newly" for clarity
L129: sequences
L134: Delete "individually for each"
L197: Define "higher incidence". Were all polymorphism lengths counted equally? Present in multiple specimens or multiple clones?
L198: helices
L208: How do polymorphic sites differ from mutation events?
L241: Change showed to suggested
L267: polymorphisms
L285: Capitalize "we"
L293: Edit for clarity here and throughout paper, what is a class vs. polymorphism vs. mutation event
L309: delete and
Figure 1: Provide the n for each taxon sequenced/cloned

Reviewer 2 ·

Basic reporting

English is OK, just a few corrections detailed below:
l. 64: “50 extant morphospecies” instead of “50 species”
l. 101-104: sentence too long
l. 114: “Briefly, all” instead of “Briefly, All”
l. 118 and 120: repetition with “resulting dataset”
l. 141-143: This sentence is a bit difficult to follow.
l. 309: “Microperforate and Non-Spinose species” instead of “Microperforate and Non-Spinose and species”
The first sentence of the abstract and the introduction are too wide, for example speaking about prokaryotes and eukayotes, when the paper focuses on a small part of one microbial group, the foraminifera. The introduction needs generally to focus more on the results actually obtained. It presents the former publications as only describing the intragenomic variability, but the present paper does not go further into understanding these mechanisms (contrary to what we could hope from the end of the introduction). On the other hand, I am missing an overview of the place of planktonic foraminifera within the whole group of forams and how representative they are for this group.
There is a missing reference for variable regions names: Pawlowski & Lecroq 2010 (l. 117)
Raw data: apparently you sequenced new data (l. 101), you need to publish these new sequences in a public database.

Experimental design

More details in the M&M section are needed. Which part of the SSU was used, the 3’ end or the complete gene? It would be good to clarify with fragment lengths (l.102). Some words about the error rate of polymerases would also be useful as you indirectly compare PHUSION and Taq in two parts of the manuscript (l. 103 and l. 295). The annotations on the secondary structure may be worth publishing. Do you show the trees presented l.178-187? If not, it would be a good idea to have them presented in supplementary material.

Validity of the findings

Traditionally in micropaleontology, planktonic and benthic forams are separated and studied by different specialists. Here, this partition is annoying and quite artificial as you want to look at the evolution of foraminifera. Planktonic forams are a polyphyletic group with different lineages issuing from one taxon of benthic forams (rotaliids as cited in the manuscript). Sometimes you speak about these lineages as clades, other times as morphogroups. What is their phylogenetic status, is each of these group monophyletic or is the grouping based on morphology?
To have a good overview of evolution in planktonic forams, it would make sense to include at least some rotaliids in this study.

Additional comments

l. 20: why speak about prokaryotes if you do not study them after?
l. 26-27: this sentence is not clear, does it mean that the number of variable regions varies?
l. 61-62: only planktonic forams, a tiny part of forams!
l. 71-72: what is the length of the fragments?
l. 74-75: and the monolamellar group?
l. 123: maybe better to use SSU instead of 18S as this gene is much longer in forams than other eukaryotes and will be heavier than 18S
l. 131: is direct sequencing useful for this type of study?

---

## Round 0.2 · Minor Revisions

I have heard back from the same two reviewers, one of whom has some minor comments that should be easy to address. As well, looking over your current version, please check this one small thing - ensure "sp." is not in italics. I anticipate being to accept a revised version soon after resubmission.

·

Basic reporting

The authors report on the intragenomic variability in planktonic foraminifera clades by comparing multiple clones of SSU rDNA from individual cells to direct sequencing products and publicly available sequences. A clear and concise introduction frames the hypotheses in this study, using recent relevant literature.

The authors have rephrased sentences that were ambiguous and edited sentences for clarity throughout.

I am satisfied with the changes that have been made and believe the manuscript is suitable for publication.

Experimental design

The research presented here falls within the scope of PeerJ and addresses critical knowledge gaps in our understanding of microbial diversity, including the possible evolutionary origins of intragenomic variability.

The authors have added details in the methods and improved their figures and tables, which now clearly describe their main findings.

Validity of the findings

The conclusions have been well-organized and supported by recent relevant literature.

Reviewer 2 ·

Basic reporting

Most of the authors' answers to my comments were satisfactory. I appreciate the fact that the authors added some benthic foraminiferal data. Concerning the 18S vs SSU, it is not because a big part of the community is using 18S for forams, that it means it is rightly used…

Experimental design

No comment

Validity of the findings

No comment

Additional comments

l. 67: it is more logical to precise 50 extant species (instead of extant clades in l. 68)
l. 143: sp. not in italic
l. 170: “cloning effort” instead of “Cloning effort”
l. 198: Aze et al. (2011) instead of (Aze et al. 2011)
l. 266 & l. 268, Fig. 4: here you speak about “benthic rotaliids”, why not use the words “benthic Rotaliida” as you use Rotaliida again later (l. 319)?

---

## Round 0.3 · accepted · Accept

Thank you for these final edits; I am happy to move this into production. Congratulations!